# HIWE: Scene Importance Weighted Encoding for Accelerated Training of Radiance Fields

## Abstract

Neural radiance fields (NeRFs) have emerged as a powerful scene representation technique to implicitly encode radiance information in space. Recent works demonstrated that using a grid-based positional encoding to encode 3D radiance information in space achieves fast training speeds, often requiring only a few minutes of training on small-scale synthetic datasets. However, training a NeRF model that uses a grid encoding on large outdoor scenes requires several hours of training. In many scenarios, large scenes may have different amounts of detailing at different regions, with reconstruction/representation quality more important for some detailing compared to others. Different regions of the scene are however given equal importance and thus typically no regions of the scene are prioritized in allocating parameters in the learned model. In this work, we propose a new grid-based positional encoding technique that integrates scene importance information in large scenes to accelerate training. Our encoding flexibly allocates more model parameters to learn the radiance information in regions of the scene that are deemed more important. This ensures that the more detailed scene regions are represented with a larger number of parameters, allowing more detailed radiance information to be encoded. With our approach, we demonstrate higher quality representation for the important parts of the scene compared to state-of-art techniques for instant NeRF training, while enabling on-par or faster training times as state-of-art NeRF models and small model sizes.

## 1 Introduction

Creating a 3D representation of various objects in a scene from a sparse set of 2D images taken from different angles, and their corresponding poses is a fundamental problem in computer graphics and computer vision with a large body of prior research. In this body of work, neural radiance fields (NeRFs) (Mildenhall et al., 2020) have emerged as a powerful scene modeling technique that has demonstrated impressive and photo-realistic rendering of the scene from novel views. NeRFs parameterize the view-dependent radiance at each point in space using a neural network. NeRFs take the 3D coordinate of a point in space as well as the view direction as input and output the radiance emitted at this point in the corresponding direction. Using a differentiable volume renderer (Kajiya & Von Herzen, 1984) to render images using the radiance field and comparing them against a set of captured training images, the NeRF model parameters can be trained with gradient optimization. NeRFs traditionally required several hours of training to capture high-quality scene details, and several recent works (Fridovich-Keil et al., 2022; Sun et al., 2022; Müller et al., 2022; Liu et al., 2020) aim to speed up NeRF training. Among these works, Instant-NGP (Müller et al., 2022) is particularly noteworthy in achieving fast training speeds, requiring just a few minutes of training on small-scale synthetic datasets.

When creating 3D representations of large-scale outdoor scenes, we may intend to capture high-quality scene details of a large expanse of space, with different sections of the scene demanding a different levels of detail. For example, a drone that surveys a large area of land might capture certain regions consisting of objects of interest close up, capturing intricate details of fine-grained features, while observing other areas from a distance where only general characteristics are recorded. As another example, regions of the scene could be identified by an upstream process, for example, via text using a language model (Kerr et al., 2023)). In other cases, some regions of the scene may simply possess more important detailing and objects than others. In these scenarios, different parts

of the scene can be assumed to have different levels of "importance" in terms of the amount of detail that should be captured.

Capturing a large scene, for example from images taken from overhead by a drone, with instant NeRF methods (such as Instant-NGP) requires a large model to encode the fine details of the scene. A number of works enabling instant NeRF training (Fridovich-Keil et al., 2022; Sun et al., 2022; Müller et al., 2022) utilize a *grid-based* positional-encoding, that translates a 3D location ($x, y, z$ spatial coordinate) into a latent space feature that is then processed by an MLP. Grid-based encoding considers a voxel grid in 3D space, in which each corner is associated with a feature vector with trainable parameters (or embeddings). The 3D position is encoded into a latent space feature vector by aggregating (interpolating) the embeddings at the corners of the voxel containing the point. As the radiance information in each voxel is exclusively encoded by the few embedding parameters in its neighborhood, these parameters can be quickly optimized to encode the radiance field within the voxel, enabling fast training. While this encoding enables fast training for smaller scenes, encoding fine-grained features in large-scale scenes requires using a high resolution voxel grid, thus requiring a large amount of memory (see Sec. 3.1). Prior works have addressed this problem by using a hash-encoding (Müller et al., 2022), or decomposed tensor representation (Chen et al., 2022) to store the voxel-grid parameters. These techniques work well for small and synthetic datasets, but capturing finer details of large scenes still requires a large model that requires long train times (several hours). This is because the training phase has to optimize a large number of model parameters and requires a large number of training samples. For instance, training InstantNGP on the `pipes1` (see Sec. 4.1) scene of the drone-deploy (Pilkington, 2022) dataset that is able to achieve peak quality requires 4-5 hours of training on a large model of over 750 MB in size.

While there are often only a few parts of the scene that are need to be captured at a higher degree of detail, the training process for these scene parts (which for example contains subjects of interest) is not prioritized. This makes generating an accurate representation of *important* locations slower. To capture a large outdoor scene, for instance, we observe that a large portion of training time is spent in optimizing the high-frequency parts of the scene such as the grass, highly textured sand, etc. (see Sec. 3.1). Hence, the training process and model used by NeRFs currently do not consider the level of importance and desired detailing for different parts of the scene, and the entire region is considered to be of equal importance in the training process.

In this work, we aim to integrate importance of different regions in a large scene within the representation encoding to accelerate training with instant NeRF methods with a novel framework, Hierarchical Importance Weighted Encoding (HIWE). To this end, we first quantify scene importance at different 3D locations by defining an *importance distribution function*, which is a probability distribution function over 3D space coordinates. This importance distribution function can be generated automatically (e.g., by using density of structure from motion points to identify surfaces) or manually where the user specifies objects of importance in the scene (Sec 3.2). HIWE uses a novel grid-based positional encoding technique which maps 3D spatial coordinates into a latent feature vector using a different number of trainable parameters to encode local regions based on the amount importance assigned to each region. HIWE uses a bounding box hierarchy to store the learned parameters instead of a regular voxel grid. Each axis aligned bounding box is associated with a fixed number of trainable parameters that encodes radiance information for the spatial region of the scene mapped to the bounding box. Bounding boxes can have different sizes. Larger bounding boxes would encode any 3D region with lower resolution (i.e., fewer parameter per unit volume) than smaller bounding boxes. Regions with less desired importance can thus be encoded by larger bounding boxes. The same region can also be mapped by multiple overlapping bounding boxes to also increase the number of parameters that are used to represent that region. By controlling the location, number, and sizes of the bounding boxes, we can flexibly provision regions of higher importance with more parameters (described in Sec. 3.3). This allocation of bounding boxes is done using the importance distribution function for the scene.

A key challenge with HIWE is to efficiently index and identify the local-grids for any point in space. To enable this lookup efficiently, we pose it as an ray-primitive intersection problem (see Sec. 3.5) in which the ray consists of a ray origin and an infinitesimal extent, and the primitive is the local grid's surrounding bounding box. We make use of the hardware-accelerated ray-primitive intersection routine to retrieve the bounding box containing the position, and implement this using the NVIDIA optix application framework (Parker et al., 2010).

This work makes the following contributions. First, we propose a novel importance-weighted positional encoding that flexibly allocates more model parameters to regions/objects that have more important parts of a large-scale scene. Second, we develop a bounding volume hierarchy-based implementation of our positional encoding that is able to leverage HW-accelerated ray-bounding box intersection primitives for fast feature indexing. Third, we demonstrate that HIWE's encoding enables higher quality representation for important regions of large-scale scenes than state-of-art methods, while still ensuring fast training times and small model sizes. We evaluate our method on a large scale outdoor dataset captured from a drone (Pilkington, 2022) and we compare against state-of-the-art NeRF techniques that use a grid based encoding techniques, both qualitatively and quantitatively (by measuring PSNR, SSIM and LPIPS (Zhang et al., 2018) metrics). With HIWE positional encoding, we demonstrate that we can achieve an increase in PSNR of up to $2.3dB$ over these methods in our evaluation, when training a model of size $< 100MB$ requiring requiring fewer than 15 minutes of training (with 30,000 iterations) on a high end GPU.

## 2 RELATED WORK

**Neural Implicit scene representations.** Neural networks can be used to parameterize information in signals (such as images, 3D scenes) as a continuous, differentiable function (Sitzmann et al., 2020; Park et al., 2019). These methods have emerged as a powerful tool to learn 3D scene representations, in which a neural network function maps a 3D spatial coordinate to the physical parameters of the scene. This enables representing object geometries/surfaces using signed distance (Li et al., 2023c; Wang et al., 2021; Yariv et al., 2020; Niemeyer et al., 2020; Chibane et al., 2020; Park et al., 2019), occupancy (Mescheder et al., 2019), material and surface properties such as reflectance/transmittance and textures (Saito et al., 2019; Henzler et al., 2020), or radiance with neural radiance fields (Mildenhall et al., 2020). Neural radiance fields map 3D spatial coordinate and a view direction to the radiance ($RGB\sigma$) emitted in the corresponding direction at that location. Several works aim to render photo-realistic novel views from the learnt model of the scene on small scale datasets (Barron et al., 2021; 2022; 2023). However, extending these methods to learn a high quality representation of large outdoor scenes requires several hours of training, with a large model. Compared to these approaches, our work aims to quickly learn scene parameters representing pre-specified, targeted regions in large scenes to enable accelerated NeRF training.

**Fast NeRF training with grid-based positional encoding.** A 3D positional encoding transforms 3D spatial coordinates into a latent space feature, which encodes local radiance information for NeRF training (Liu et al., 2020). Grid-based positional encoding divides the 3D space using a grid (a voxel grid), and associates an embedding vector to each corner of the grid. The latent feature vector is computed by aggregating the embedding associated with the corners of the voxel containing the coordinate. Grid-based encoding is a commonly used encoding technique to enable fast training, demonstrated by works such as Fridovich-Keil et al. (2022); Sun et al. (2022); Yu et al. (2021); Liu et al. (2020). However, these methods require large amounts of memory to represent large scenes, as it requires allocation of a large high-resolution voxel grid. Several approaches have been proposed to address this problem, such as using a small array indexed by a hash function to store the parameters (Müller et al., 2022), or representing the volume as a 4D tensor of embedding vectors and parameterizing it with its low-rank factors (Chen et al., 2022). However, these works require significantly high training times for large outdoor scenes, as a large number of grid encoding parameters are still required to capture all details of the scene (see Sec. 3). We compare HIWE to other grid-encoded representation techniques in Sec. 4.

**Large scene representations with NeRFs** Representing large scenes, such as a city-scale scene captured from an overhead drone was investigated by several works. Prior works Xiangli et al. (2022); Xu et al. (2023) propose special model architectures to effectively represent multi-scale data captured at different resolutions. Other works Turki et al. (2022); Tancik et al. (2022); Zhang et al. (2023) propose using several smaller NeRF models to represent a large scene, with each model representing a segmented region in space. Similarly, F2Nerf (Wang et al., 2023) uses a spatially subdivision regions of space and learns implicit grid encodings on individual grids based on the number of samples captured by the camera. While these approaches enable high quality training of large outdoor scenes, achieving a high quality representation requires several hours of training. On the other hand, we are concerned with training a small model ($< 100MB$) with reconstructing an accurate scene representation with than 15 minutes of training time.

**Compression techniques for NeRFs.** In order to reduce the storage size of the scene representation with minimal loss in quality, several works propose techniques such as compressed encoding based on the level of scene detail (Martel et al., 2021; Lindell et al., 2022), pruning the parameters that encode radiance in empty space (Liu et al., 2020), using a lower floating point precision for less detailed scene parts (Li et al., 2023a) or embedding features at the corners of an octree representation to represent different level of detail (Yu et al., 2021). CC-Nerf (Tang et al., 2022) computes a low-rank decomposition of the dense tensor to obtain an efficient scene representation. While our training method relies on using a smaller model for fast training, compression methods often require long processing times as they either require a post-processing step or require several iterations of simultaneously training and pruning of the model. HollowNerf (Xie et al., 2023) observes the limitation of needing a large model to obtain a high quality reconstruction on synthetic datasets. They show that by not learning radiance at the interior (unseen) parts of the objects, they could train a light-weight model. However, this approach does not necessarily lead to faster training, and this observations was only demonstrated on smaller synthetic datasets.

**Efficient training sampling techniques for faster convergence.** NeRF training optimizes a loss defined as the difference between the RGB values of the pixels in the rendered image to the corresponding pixels of the images in the training dataset. To speed up convergence, sampling pixels for training weighted by the loss contributed by each pixel is implemented in InstantNGP (Müller et al., 2022; Li et al., 2023b). When training large outdoor scenes, however, a significant portion of training resources are allocated to less important, high frequency details of the scene (see Sec. 3). We propose a new pixel sampling method that prioritizes choosing the important scene parts instead.

**Other radiance field representation techniques.** Explicit scene representation techniques model the scene with a collection of several parameterized geometric primitives (like point clouds or meshes). Gradient based optimization methods that use a differentiable rasterizer (Laine et al., 2020) are increasingly becoming popular for inverse rendering problems to learn the parameters of these representations primitives efficiently. ADOP Rückert et al. (2022) uses an efficient differentiable point cloud rasterizer to learns a point cloud representation of the scene. 3D gaussian splatting (Kerbl et al., 2023) is a recent work that represents a scene with a set of 3D gaussians, where each gaussian is parameterized by its 3D position, covariance, view dependent radiance information and transparency $\alpha$. The gaussian parameters can be learnt by gradient descent optimization on the loss between the ground truth images and the rendered images. Rendering of the 3D gaussians is done by $\alpha$ blending using a differentiable rasterizer. The fast algorithm used in differentiable rasterization (Lassner & Zollhofer, 2021) enables rendering and learning to proceed in a very fast manner. See Sec. 4.4 for a more detailed discussion in comparison to our work.

## 3 METHOD

A typical NeRF network architecture consists of a 3D positional encoder, a directional encoder (typically the spherical harmonic (SH) coefficients of the view direction (Fridovich-Keil et al., 2022)) and a multi-layer perceptron (MLP). The positional encoder takes a 3D coordinate $(x, y, z)$ as input and produces a latent space feature vector. This latent space feature along with the SH coefficients is passed as input to the MLP to generate the radiance (density $\sigma$ and the $RGB$ color). Rapid training of NeRFs is achieved by using a *grid-based* positional encoder (Liu et al., 2020; Fridovich-Keil et al., 2022; Sun et al., 2022). A grid-based encoder considers the 3D space discretized into a voxel grid, and each grid corner is associated with an embedding vector. The latent feature vector of a 3D point is derived by interpolating the embeddings at the corners of the voxel containing the point.

### 3.1 TRAINING OF HIGH-FREQUENCY SCENE REGIONS

Fast training is possible with a grid based encoding because the parameters at the grid corners encode the radiance information of only the neighbouring voxels. This small set of parameters (along with that of the small MLP) can be quickly optimized for each voxel leading to fast training (Liu et al., 2020; Müller et al., 2022). However, capturing large outdoor scenes in detail requires the parameterization of radiance inside each voxel with a high-resolution grid. Updating and optimizing each parameter requires a large amount of samples and long training times. Fig. 1 shows large scene captured from a drone, which is a part of the house2 dataset (see Sec. 4.1). On using a smaller model to capture the scene (an 8-level multi-resolution hash encoding model with a maximum resolution grid of size of 2048 voxels spanning the extent of the scene, and a hash-encoding size of $2^{19}$ elements per level), we find that the model is inadequate to represent all parts of the scene at high

enough quality. However, a lager model (16 levels with a hash table size of $2^{22}$), is able to learn a more detailed scene representation but requires hours of training (200k iterations). The training process for the large model spends a significant proportion of the time to learning the high-frequency regions of the scene (in this instance, the grass).

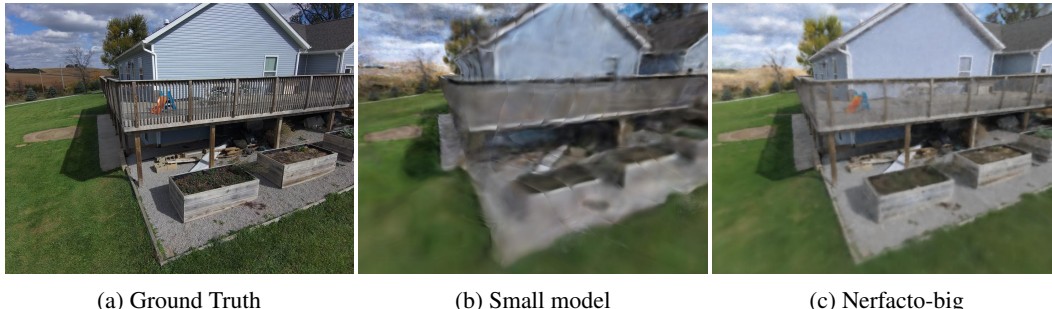

(a) Ground Truth         (b) Small model         (c) Nerfacto-big

Figure 1: Training a large scene with nerfacto - the smaller model seen in (b) is not able to capture scene adequately. Using a large model is able to significantly improve on the representation. However, using a larger model in (c) results in better quality, but requires longer training times.

When using InstantNGP, the large number of parameters for this high frequency region contributes significantly to hash collisions, leading to lower quality. Hence, encoding the region with fewer parameters which can be made to quickly converge is a more efficient way to encode these regions. In general, these scene components can be considered to be *less important*, as they add little semantic or structural information and are challenging to be represented efficiently. This importance information is generated prior to training as described in Sec. 3.2. We leverage this information to more efficiently allocate model parameters to different regions of the scene. An overview of our approach is depicted in Fig. 2.

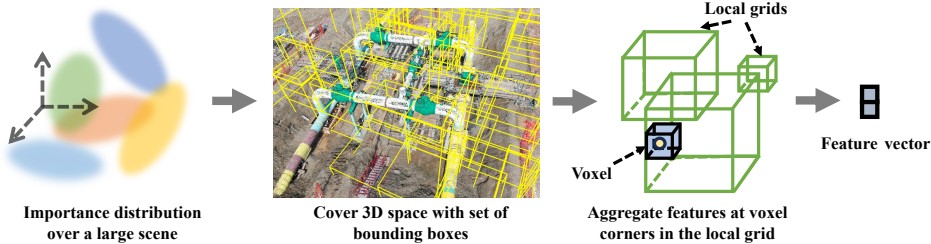

**Importance distribution over a large scene**     **Cover 3D space with set of bounding boxes**     **Aggregate features at voxel corners in the local grid**

Figure 2: HIWE importance weighted positional encoding overview

### 3.2 IMPORTANCE DISTRIBUTION IN THE SCENE

In order to describe the importance of different parts of the scene, we define an importance distribution function $f : R^3 \rightarrow R$. $f$ is a probability distribution function defined over 3D space which describes the amount of importance that should be given to each location in space. From the definition of a probability distribution, we can describe the importance of a volume $V$ as the integral of $f$ over volume V (where $d\tau$ is the volume element).

$$Importance(V) = \int_V f(x, y, z)d\tau \tag{1}$$

The importance distribution for any scene can be determined and configured based on the use case, either manually or automatically. For example, there may be a particular scene component that is of specific interest such as a statue in the middle of a field of grass. In this scenario, the statue would be assumed to be the more important scene component, requiring more representation detail. In this work, we consider two potential scenarios: **(i)** A single region or object in a scene is defined as important. We describe the importance distribution for this case with a 3D gaussian distribution centered at the position to focus on that region or object (see Sec. 4.3). **(ii)** Regions of the scene close to object surfaces are marked as important. A reasonable candidate used to automatically mark the region around the surfaces as important is the normalized density of the structure-from-motion (SfM) point cloud (see Sec. 4.2).

### 3.3 IMPORTANCE WEIGHTED POSITIONAL ENCODING ARCHITECTURE

**Encoding local scene features with local grids.** We propose a new grid-based positional encoding technique, HIWE, which maps 3D spatial coordinates to a latent feature vector using a different number of trainable parameters to encode local regions based on the amount importance assigned to this region. HIWE uses a bounding box hierarchy to store the learned parameters instead of a voxel grid. Each axis aligned bounding box is associated with a local grid comprising of $N \times N \times N$ voxels (where $N$ is a fixed hyperparameter) with each voxel corner associated with trainable parameters. Each local grid encodes radiance information of scene elements in the region enclosed within the bounding box. Each bounding box can be mapped to a 3D region of any size but would still have the same sized local grid (i.e. $N \times N \times N$ voxels). Thus, larger bounding boxes would encode any 3D region with lower resolution (i.e., fewer parameter per unit volume) than smaller bounding boxes. The same region can also be mapped by multiple overlapping bounding boxes to also increase the number of parameters that are used to represent that region. Regions with less desired importance can thus be encoded by larger bounding boxes. By controlling the location, number, and sizes of the bounding boxes, we can flexibly provision regions of higher importance with more parameters. We describe how the bounding box allocation is done in Sec. 3.3. The feature vector associated with any 3D point $(x, y, z)$ is generated by aggregating (linearly interpolating) the 8 embedding vectors at the corners of the local grid containing the point, as shown in Eq. 2. The feature vector for a 3D point when it maps to multiple overlapping local grids is determined by calculating the arithmetic mean of the those computed from each individual local grid (shown in Eq. 3).

$$fv_{local}(x, y, z) = Interp_{x,y,z \in bbox}(x, y, z) \tag{2}$$

Where $fv_{local}$ is the feature computed by the local grid

$$feature\_vector(x, y, z) = \frac{1}{N_{bboxes}} \sum_{bbox \in bboxes} fv_{local}(x, y, z) \tag{3}$$

**Generation of local grids according to scene importance.** At the start of training, our encoding allocates a set of bounding boxes, each corresponding to a local grid. Fig. 3 shows the procedure for generating bounding boxes. The size of the bounding box is proportional to the desired importance of any given region. The total number of bounding boxes ($N_{bbox}$) is fixed and is configured ahead of time, and decides the total number of learned parameters in the model. To determine the center of each of the $N_{bbox}$ bounding boxes (i.e., the 3D location that it maps to), we sample $N_{bbox}$ points from the importance distribution as shown in Fig. 3. Thus, the important regions of the scene are allocated more bounding boxes. The size of each bounding box is inversely proportional to the deemed importance of the region around the center of the bounding box. To determine this deemed importance in the surrounding region, we again sample the original importance distribution to generate a point cloud with a large number of sampled points. Each bounding box is then sized such that they all have the same number of sampled points from the point cloud (hyperparameter $N_{p0}$). Thus regions with higher importance will have smaller bounding boxes (as the density of sampled points is higher). The less important regions will automatically be sized to be bigger as those regions have a lower point density.

$$Sz_{bbox} = \beta * (\texttt{cube\_enclosing\_points}(N_{p0}, c_{bbox}))^{.33} \tag{4}$$

Where $Sz_{bbox}$ is the size of the bounding box, $N_{p0}$ is the number of points to be enclosed within the bounding box, $\beta$ is a hyperparameter (grid-size constant), $c_{bbox}$ is the bounding box center and `cube_enclosing_points` returns minimum volume of a cube centered at $c_{bbox}$ that encloses $N_{p0}$ points. This process is done $L$ number of times, each time with a different $N_p$ (i.e., points per bounding box). Please refer to Appendix A for details. This generates a hierarchy of bounding boxes to represent the scene at different scales of resolution. In our evaluation, $L$ was set to 8.

### 3.4 IMPORTANCE WEIGHTED PIXEL SAMPLER

For each training iteration, assigning a higher weight in selecting pixels that render scene regions with higher importance enables faster training around the important parts of the scene. Hence, we implement an importance weighted pixel sampler that chooses pixels weighted on the amount of importance seen by each pixel. From a ray $r(t) = o + td$ casted from the camera center to the image, where $o$ represents the ray origin, $d$ is the ray direction and $t > 0$ is the time parameter, we can evaluate the amount of importance seen by each pixel as:

$$Importance_{pixel}(p_x, p_y) = \int_{tn}^{tf} T(t)f(o + td)dt \tag{5}$$

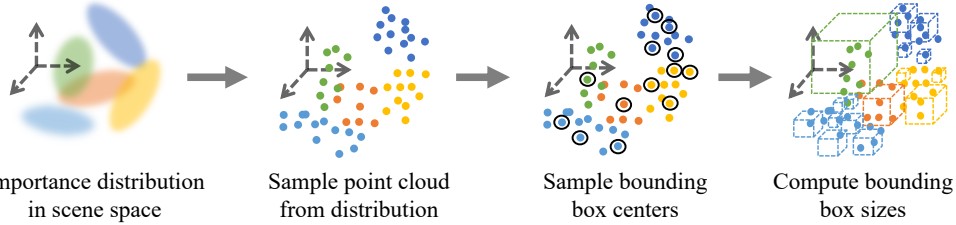

Figure 3: **Generation of bbox features:** Starting from the importance distribution function, we sample a point cloud in 3D space from the distribution. The bounding box centers are sampled from this point cloud distribution uniformly. From these bounding box centers, the size of each box is computed such that it encloses a fixed number of points of the point cloud.

$$T(t) = exp(-\int_t^{tf} \sigma(r(t))dt) \tag{6}$$

where $f$ is the importance distribution function, and $\sigma$ is the volume density, as defined when for volume rendering (Kajiya & Von Herzen, 1984). Evaluating this quantity is not trivial. However, a simple approximation is to consider the space being constituted by regions where either $\sigma = 0$ (unoccupied regions in space), or $\sigma(t) = \delta(t - t_0)$ at regions where there is occupancy. Here, $t_0$ is the ray's time at which it hits the point of occupancy. In this scenario, the importance seen by pixel is given by $f(r + t_0 d)$. From Sec. 3.3, we know that the volume of the bounding box seen by the corresponding ray is proportional to the importance of the region. Hence, the pixel's importance is proportional to the volume of the first local grid bounding box encountered by the pixel. In our implementation, we take the the lowest hierarchy level of the HIWE local grid bounding box intersected by the ray cast by the pixel to approximate importance seen by each pixel.

### 3.5 FAST INDEXING OF BOUNDING BOXES

An important challenge in implementing HIWE positional encoding is indexing the learned parameters corresponding to each point in space. This is challenging because we need to identify all the bounding boxes that enclose each 3D coordinate. From our experiments, the number of bounding boxes ($L * N_{bbox}$) allocated is about 10,000 to 100,000 for large outdoor scenes and corresponding bounding boxes need to be identified for all the training points (approximately 100K). A brute force search across all bounding boxes for each point is infeasible. The problem of finding enclosing bounding boxes for a large batch of points can be posed as a ray-bounding box intersection problem, with the ray's maxmimum time extent is limited to 0.0. Batched ray-bounding box intersection finding operations can be efficiently carried out using a bounding-volume-hierarchy tree search built on top of the HIWE bounding boxes. By leveraging hardware acceleration support for ray-bounding box intersection tests using libraries such as NVIDIA-optix (Parker et al., 2010) or Intel-embree (Wald et al., 2014), we develop an efficient approach to index local grids from a batch of 3D point samples.

## 4 RESULTS

### 4.1 EXPERIMENTAL SETUP

We implement our approach as a new method in nerfstudio (Tancik et al., 2023), a standardized framework to develop and evaluate NeRF models. We demonstrate our approach on large scale drone-deploy dataset (Pilkington, 2022) which consists of sequences of high-resolution images data captured by a drone on large scale scenes for the following data sequences: `pipes1`, `tower1`, `tower2`, `house1`, `house2`, `house3`, `ruins1`, `ruins3`. We generate the poses used for training and evaluation using colmap (Schönberger & Frahm, 2016; Schönberger et al., 2016). We evaluate all our trained models on a desktop computer with an NVIDIA RTX4090 GPU for 30k training iterations, which takes about 15 minutes of training time. We compare our approach with the baseline nerfacto model (Tancik et al., 2023), which uses 16 levels of resolution, max-resolution voxel grid size of 2048 and the hash table size set to $2^{19}$. We also present results for nerfacto-h22, which uses a hash table size $2^{22}$, the nerfacto-big model (Tancik et al., 2023) and TensoRF (Chen et al., 2022). We demonstrate our approach in 2 scenarios: **(1)** When a region's importance is determined by the density of structure from motion point cloud of the scene; and **(2)** One specific region of the scene is marked as important by the user.

## 4.2 Scenario 1: Importance determined by SfM point cloud density

In this scenario, we define the importance of the scene as being proportional to the density of the sparse structure from motion (SfM) point cloud. We sample from this distribution to generate 8 levels in the bounding box hierarchy ($L$) with a grid size constant $\beta = 1.1$.

Fig. 4 shows a qualitative evaluation of our approach. We observe that our model is able to produce visibly sharper images reconstructions of detailed scene parts using the same number of parameters after the same number of training iterations. However, the `house3` data sequence has a particularly degraded quality, with floaters present in the scene. We find that this was due to improper pose information produced by colmap that can be addressed by training for a few more iterations. Table 1 shows the LPIPS, SSIM and PSNR measured for each of the scenes in the dataset. We see better accuracy in the images rendered by our approach compared to others.

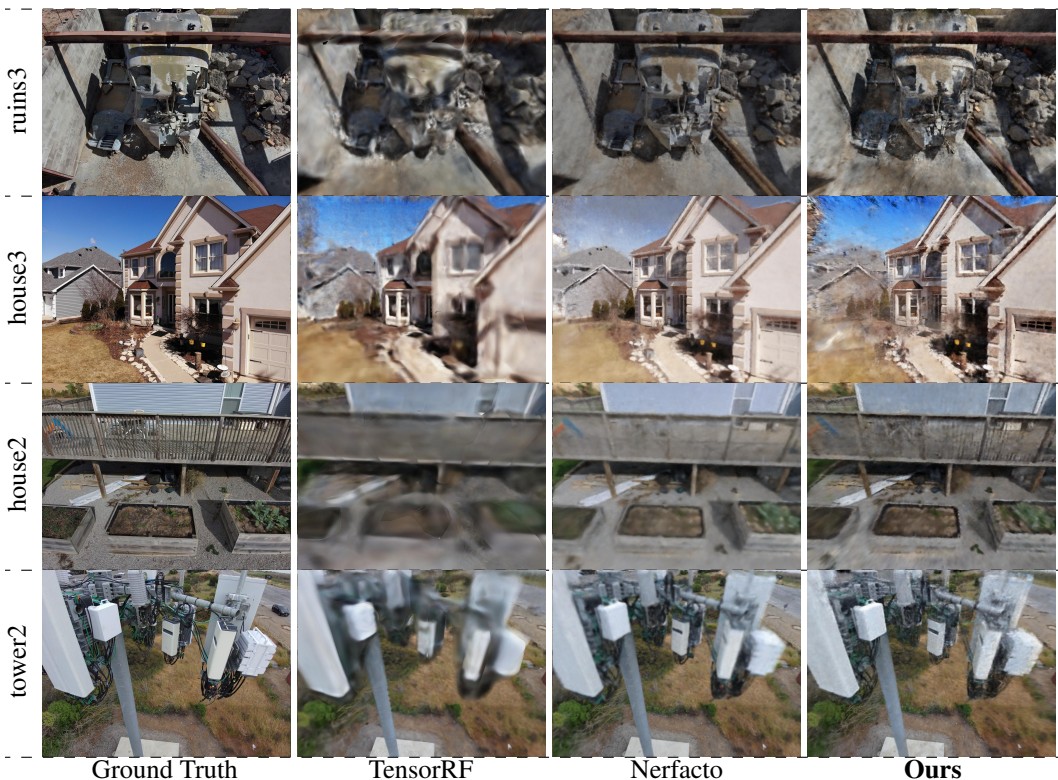

Figure 4: Visual comparison of scenes rendered with various NeRF methods.

## 4.3 Scenario 2: Single region marked as important

In this scenario, we mark a specific region of the scene as important with a standard normal gaussian distribution function around the region to describe the scene importance. Fig. 5 shows the rendered image reconstructed with the ground truth representation. We set the important region around the center of the graffiti shown in the figure. We observe with only 30K iterations of training, our model is able to recover details of the particular region with greater quality compared to the nerfacto-big model trained for 100k iterations.

## 4.4 Comparison to Scene Representation with 3D Gaussian Splatting

While are work targets neural radiance fields, we also compare against another non-neural implicit radiance method, 3D gaussian splatting (3DGS) (Kerbl et al., 2023) that enabled high speed reconstruction using 3D gaussians. We find that individual gaussians are able to accurately encode high frequency scene regions efficiently with short training times (about 30 minutes), and is able to achieve a better rendering quality than NeRF-based methods. Table 2 shows a comparison of our approach with 3DGS after training for 30K iterations. However, a fair and direct comparison between these fundamentally different implicit radiance fields methods is challenging as each representation

| | pipes1 | tower1 | tower2 | house1 | house2 | house3 | ruins1 | ruins3 |
|---|---|---|---|---|---|---|---|---|
| **PSNR↑** | | | | | | | | |
| **Ours** | 23.1 | 19.1 | 21.64 | 18.51 | 20.86 | 17.35 | 17.5 | 18.25 |
| nerfacto | 21.59 | 17.75 | 19.88 | 17.15 | 19.94 | 17.83 | 18.15 | 18.07 |
| nefacto-h22 | 21.11 | 17.9 | 19.94 | 16.94 | 19.91 | 17.99 | 17.9 | 18.1 |
| nerfacto-big | 22.5 | 17.7 | 19.57 | 17.75 | 17.6 | 12.16 | 18 | 18.08 |
| TensoRF | 18.26 | 16.97 | 20.9 | 17.66 | 20.65 | 16.72 | 17.41 | 18.31 |
| **SSIM↑** | | | | | | | | |
| **Ours** | 0.7 | 0.45 | 0.55 | 0.57 | 0.38 | 0.56 | 0.38 | 0.42 |
| nerfacto | 0.62 | 0.47 | 0.46 | 0.52 | 0.38 | 0.45 | 0.39 | 0.44 |
| nefacto-h22 | 0.62 | 0.47 | 0.46 | 0.47 | 0.38 | 0.46 | 0.4 | 0.45 |
| nerfacto-big | 0.72 | 0.42 | 0.43 | 0.57 | 0.38 | 0.45 | 0.41 | 0.44 |
| TensoRF | 0.36 | 0.44 | 0.44 | 0.51 | 0.36 | 0.41 | 0.44 | 0.32 |
| **LPIPS↓** | | | | | | | | |
| **Ours** | 0.20 | 0.7 | 0.61 | 0.47 | 0.74 | 0.60 | 0.63 | 0.7 |
| nerfacto | 0.31 | 0.72 | 0.69 | 0.45 | 0.80 | 0.57 | 0.62 | 0.73 |
| nefacto-h22 | 0.29 | 0.65 | 0.69 | 0.42 | 0.80 | 0.56 | 0.62 | 0.7 |
| nerfacto-big | 0.19 | 0.68 | 0.63 | 0.34 | 0.79 | 0.54 | 0.58 | 0.68 |
| TensoRF | 0.64 | 0.83 | 0.72 | 0.62 | 0.86 | 0.69 | 0.8 | 0.8 |

Table 1: LPIPS, PSNR and SSIM computed on evaluation images rendered from various models

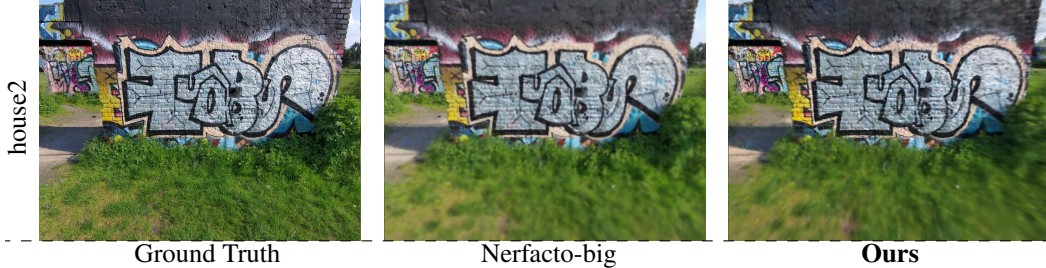

Ground Truth       Nerfacto-big       **Ours**

Figure 5: **Visual comparison between nerfacto-big baseline vs our approach, with the area around the graffiti is assigned a higher level of importance.** The definition of each individual bricks is higher around the graffitti region. The quality diminishes in the grass region and the peripheral regions of the image compared to nerfacto-big.

| | pipes1 | tower1 | tower2 | house1 | house2 | house3 | ruins1 | ruins3 |
|---|---|---|---|---|---|---|---|---|
| 3DGS | 27.01 | 22.15 | 24.2 | 27.00 | 23.52 | 22.15 | 24.66 | 24.23 |
| Ours | 23.5 | 19.10 | 21.64 | 18.51 | 20.86 | 17.35 | 17.5 | 18.25 |

Table 2: PSNR of 3DGS scene representation with our approach

has different tradeoffs. For instance, modelling the scenes as explicit gaussians typically requires a large amount of gaussian primitives to encode the scene resulting in significantly higher overhead for storage of the model (over 1GB on average vs 86MB for our model), especially for large scenes. Surface representation is also more challenging with these methods. Additionally, 3DGS relies on rasterization for fast training speeds. The rendering technique (rasterization) used will not be able to model scattering, reflective and refractive media present in the representation.

## 5 CONCLUSION

We introduce HIWE, a novel positional encoding technique for neural radiance fields that leverages the knowledge of importance in representing different scene parts of large-scale scene to accelerate training and enable higher representation quality. HIWE flexibly allocates more model parameters to encode regions of the scene with more detailing and importance. We develop an efficient bounding box-based implementation of HIWE that leverages hardware accelerated ray-bounding box intersections for fast feature indexing. With HIWE, we demonstrate better rendering quality for important regions of large scale scenes, while using similar model sizes and enabling on-par or faster training times as state-of-art NeRF methods.

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

## A  ALLOCATION OF THE BOUNDING BOXES

Here we describe here the algorithm we use to allocate the BBOX levels for our encoding. We start with the importance distribution $f$ and hyperparameters describing the bounding box generation process to produce an array of sets of bounding boxes $\mathcal{B}$. The algorihtm is shown in Alg. 1.

**Input** : Number of levels $L$, base level point limit $N_{p0}$, importance distribution $f$, number of
          bbox at level 0, $N_{bbox_{baselevel}}$, growth factor $b$, grid constant $\beta$

**Def.**   : · cube_enclosing_box($c, N_e, \mathcal{V}$): returns minimum volume cube centered at $c$ that
          encloses $N_e$ points from set $\mathcal{V}$.
          · sample_points($p, num_p$): Sample $num_p$ points from a probability distribution $p$

BBox array $\mathcal{B} = \{\}$
$N_p \leftarrow N_{p0}$
$DIST_{MAX} \leftarrow 100000$
Distribution point cloud $\mathcal{P} \leftarrow$ sample_points($f, DIST_{MAX}$)
**for** $lvl \leftarrow 1 ... L$ **do**
    BBOX Centers $\mathcal{C} \leftarrow$ sample_points($f, N_{bbox}$)
    BBOX Extends $\mathcal{E} \leftarrow \{\}$
    **for** $c \in \mathcal{C}$ **do**
        $vol_{bbox} \leftarrow$
        cube_enclosing_box($c, N_p, \mathcal{P}$)
        $sz_{bbox} \leftarrow \beta * vol_{bbox}^{0.33}$
        $\mathcal{E} \leftarrow \mathcal{E} \cup sz_{bbox}$
    **end**
    $N_p \leftarrow N_p * b$
    $N_{bbox} \leftarrow N_{bbox}/b^{.33}$
    $\mathcal{B} \leftarrow \mathcal{B} \cup (lvl, (\mathcal{C}, \mathcal{E}))$; /* Add the level, centers, extends of BBox */
**end**
return $\mathcal{B}$;

**Algorithm 1:** Our BBOX hierarchy heneration algorithm

