# OpenReview forum: "HIWE: Scene Importance Weighted Encoding For Fast Neural Radiance Field Training"
_ICLR.cc/2024/Conference — Submitted to ICLR 2024_

### Official Review · Reviewer_zZ4B · 2023-10-31

**Soundness:** 2 fair
**Presentation:** 3 good
**Contribution:** 2 fair
**Rating:** 3
**Confidence:** 5

**Summary:**

The proposed HIWE is a grid-baed nerf method, which flexibly allocates more model parameters to important regions. First of all, an important distribution is acquired by either SFM or by manual input. The second step is to generate axis-aligned bounding boxes by weight sampling strategy. Each bbox corresponds to a local grid, where each corner is associated with a feature vector. The third step is to train a grid-based nerf, with the difference that typical grid-based nerf works on a uniform voxel grid, while the HIWE works on a set of bboxes. Experiments demonstrates a good balance between quality and training speed.

**Strengths:**

1. It is plausible to take region importance into consideration when working with nerf. The idea is simple and obviously effective - the scene is represented by a set of bboxes instead of a uniform voxel grid.
2. The nerf is forced to focus on the bboxes, especially the samller bboxes, which leads to a fast training of 15min.

**Weaknesses:**

1. It is questionable to represent the scene with bboxes.
(a) Indeed the bbox design enables flexible parameter density. However, The proposed bbox generation method is lack of control on the parameter density, i.e., we can't precisely define the parameter density / grid density at a specific region. For example, (i) The bbox centers are sampled from the importance distribution, which means it is likely that there is no bbox ar all for a region with lower (not zero) importance. (ii) In regions of higher importance, there may be lots of overlapping bbox, which is a waste of compute.
(b) A simple alternative, is to represent the scene with Octree or similar datastructure. With Octree, we can precisely control the tree depth at each region, which the problem of (1a) and (1b). Justification is necessary to clarify why bbox not Octree.

2. The acquisition of importance distribution is non-trivial.
(a) The user have to reconstruct the scene first so that the important region can be manually defined, which becomes a chicken-and-egg problem.
(b) Although the author proposed the "normalized density of the SFM point cloud", but that may not work in real world. SFM keypoints are usually on texture-rich regions, but "texture-rich" is NOT necessarily more important.
(c) The 15min training time probably does not include the generation process of the importance distribution, which seems to be unfair.

3. Inadequate experiment.
(a) There are only 2 baselines, nerfacto and TensorRG, while there are lots of other grid-based nerf method.
(b) In Table 1, the LPIPS metric is worse than baselines. However, LPIPS is usually more important than SSIM and PSNR, because SSIM and PNSR is just a metric on the statistics over the color distribution, while LPIPS is more capable of modeling the similarity of visual perception.
(c) Training time of baselines are not presented. Although the proposed method is trined with only 30K iterations, it does not necessarily mean it is fast, because Section 3.5 Indexing of Bounding Boxes can be slow. This is related to Weakness 1 - why not Octree?

**Questions:**

See weakness. Additionally, the proposed method is lack of novelty. The core idea is to represent the scene with bboxes, which is a marginal improvement over grid-based nerf methods.

---

### Official Review · Reviewer_8RFz · 2023-10-31

**Soundness:** 1 poor
**Presentation:** 3 good
**Contribution:** 2 fair
**Rating:** 1
**Confidence:** 4

**Summary:**

This work investigates the importance weights of different regions and focuses on more important regions for NeRF optimization. They propose a new grid-based positional encoding to accelerate training.

**Strengths:**

The paper first raises an observation of NeRF optimization at different levels of granularity and proposes a method using hierarchical bounding boxes to compose importance factors of different regions. I agree the intuition is reasonable to assign an importance region in a scene but I have different opinions on how this work design regions of importance.

**Weaknesses:**

1. Since HIWE assigns a fixed N^3 voxel grid for each object, the importance strategy may deal with each object with the same level of complexity regardless of whether it is big or small objects. The sentence "Regions with less desired importance can thus be encoded by larger bounding boxes" indicates larger objects are coarser. However, larger objects may also include higher complexity such as a building with textural exterior appearances or a large statue. I think the intuition behind, locating different importance regions, sounds reasonable, but the strategy here may only benefit scenes with small and delicate objects and large repetitive background. All the demonstrated results fall into this category.

2. One of the advantages is to save computation for repetitive textured regions. The detector seems to allow highly overlapping bbox, and there might be larger computation overhead to optimize voxels in repetitive space. For example, in Fig. 2 the repetitive boxes on pipes may cause large overhead.

3. The work thinks repetitive background regions are not that important, but those details actually matter for visual quality. I personally wouldn't think Fig. 5's blurriness on grass or this kind of repetitive region is non-important.

4. All the PSNR and SSIM in Table 1 are incredibly low. PSNR < 20 seems not yet converged and can only show very blurred images. The experiments are not convincing at all.

5. Including more results on standard NeRF datasets with video on multi-view can show its applicability on different scenes.

**Questions:**

See Weakness

---

### Official Review · Reviewer_BBC4 · 2023-11-01

**Soundness:** 2 fair
**Presentation:** 1 poor
**Contribution:** 2 fair
**Rating:** 3
**Confidence:** 4

**Summary:**

In this paper, the authors propose a novel grid definition to improve the performance of instant-NGP like methods on very large scene. Instead of using a uniform grid like in instant-NGP, the authors propose to derive overlapping grids sampled in regions of importance (using two possible definitions of importance, either user defined with a Gaussian density or defined according to the initial density estimated from structure-from-mention). This is then used to model large scenes acquired using drones.

**Strengths:**

Being able to render accurately large regions is extremely important. Being able to do so in a few minutes is even more so. The proposed method aims to solve both these problems at once and as such could have a strong practical impact.

**Weaknesses:**

It seems that the related work section is all over the place, even talking about compression techniques for NeRFs. This is a problem because the author don't discuss important related work and especially those that consider smart voxels structures. From the top of my head {1} and related methods adapt the concept of voxels to have a better splitting of the space using octrees for example. It is therefore very important to compare these very related ideas and explain why the overlapping grids proposed here is more appropriate than these size adaptive voxels decomposition. Note that some of these methods, like DVGO {2} or its improved version are also quite fast so could warrant an experimental comparison as well. Another related paper, thus more recent, is {3}.

The importance weighted pixel sampler is not clear in my opinion. The per pixel importance depends on the output of nerf (dependency on $T$, therefore a dependency on $\sigma$ and $t_0$). When is it then computed? Does it requires a first "importance generation" before the actual training step everytime? Or every couple of steps? Wouldn't that be counter productive since it would improve heavily the computational cost. This part should be clarified. I mention here a potential typo "the pixel's importance is proportional to the volume of the ..." should be "inversely proportional".

The impact of the importance sampling step is not clear to me. The authors mention the case of single Gaussian to define a region of interest. I'm confused why the proposed scheme would be more interesting than a NeRF 360 that focuses on the region of interest. Indeed, the content of the highlighted region would be well reconstructed while the rest would be projected into the background. In my opinion this case should therefore be considered in the experimental section (or clearly justified inside the paper).Similarly, I think that the reasoning behind the second sampling should detailed.

I also find the presentation of the results is quire poor. First, I would expect averages and highlighting of the best results in Tables to help the reader understand the best methods. I also don't think that a table is best way of presenting the results. In my opinion, a better way of visualizing the results would have been with convergence plots (for example PSNR vs training iters) instead of choosing a specific stop moment. Indeed, given how close the results are between the methods, it would make it possible to better compare the methods: maybe one achieve its final regime much more quickly than the others, for example with a couple more iterations one method will drastically improve while the other would stagnate. I expected comparisons with more relevant methods as well. Indeed, the method comapres itself to three different sizes of the nerfacto model and TensoRF. Where are the more appropriate methods (with some mentioned in the related section actually)? In particular, one could list zip-nerf or tiling the scene with multiple instant-NGP networks. It is not clear either what nerfacto model was used for Fig. 4.

I would have also expected a more indepth analysis of the additional properties of the methods. Indeed, the proposed method is clearly outperformed in Table 2 but the authors downplay this results by explaining that the rendering technique is limited (one could point out that the "first intersection" hypothesis of the method is also quite restricting) but no such examples are shown. This also mention that surface representation is more challenging, except that no results are shown with proposed method showing that surface representation is indeed good and usable.

{1} Yu, A., Li, R., Tancik, M., Li, H., Ng, R., & Kanazawa, A. (2021). Plenoctrees for real-time rendering of neural radiance fields. In Proceedings of the IEEE/CVF International Conference on Computer Vision (pp. 5752-5761).

{2} Sun, C., Sun, M., & Chen, H. T. (2022). Direct voxel grid optimization: Super-fast convergence for radiance fields reconstruction. In Proceedings of the IEEE/CVF Conference on Computer Vision and Pattern Recognition (pp. 5459-5469).

{3} Kulhanek, J., & Sattler, T. (2023) Tetra-NeRF: Representing Neural Radiance Fields Using Tetrahedra.  In Proceedings of the IEEE/CVF International Conference on Computer Vision (ICCV)  (pp. 18458-18469)

**Questions:**

See the different remarks in "Weaknesses".

---

### Official Review · Reviewer_etAT · 2023-11-02

**Soundness:** 2 fair
**Presentation:** 2 fair
**Contribution:** 2 fair
**Rating:** 5
**Confidence:** 5

**Summary:**

This work, HIWE, tackles novel-view synthesis task for large scale scenes. HIWE alloclates voxel grids using bounding volume hierarchy. The hierarchy structure is generated ahead of the optimization by using SfM sparse point cloud or user interaction. As a result, the model is much smaller, training faster, and show comparable quantitative results.

**Strengths:**

The method of allocating more model capacity to the more complex scene part is straightforward and making senses.

**Weaknesses:**

The bounding box allocation is decide ahead of the optimization and rely heavily on SfM solver. In case that SfM fails to recover point cloud such as reflective, translucent, or scene part that is complex but observed by few views, the proposed method can't adaptively allocate more resorce for it. I think this is one of the reason that the results are much worse than the discussed 3D gaussian splatting in Sec4.4 as 3DGS adaptively growth and prune the primitives.

Missing discussion of some primitive primivie mixure methods:
- [Mixture of Volumetric Primitives for Efficient Neural Rendering, LOMBARDI et al., ToG 2021]
- [Progressively Optimized Local Radiance Fields for Robust View Synthesis, Meuleman et al., CVPR 2023]

**Questions:**

1. The implementation of fast ray-bounding volume hierarchy intersection is claimed to be one of the main contruction (Sec.1 last paragraph). The algorithm is already implementmented by the used package optix or embree, so what would be the new merit by this work?
2. How to ensure bounding volume coverage for textureless region (e.g., sky in Fig4) which SfM may not produce points?
3. What would be the upper limited of current NeRF-based SOTA on the tested dataset? As discussed in Sec4.4, 3D gaussian splatting can achieve superior results on the tested dataset but could still have some limitation due to rasterization, it would also be good to know the current best results of large scene NeRF discussed in Sec.2.
4. Paper proofread:
    - The number results of the `pipes1' scene in Table 1 and 2 are not consistent.
    - Ths SSIM of `tower2' is incorrectly highlighted (0.46 is highlighted but 0.55 is better).
    - Duplicated "requiring" in the last sentence of Sec1.

---

### Meta-Review · Area_Chair_EfrW · 2023-12-04

**Metareview:**

This paper proposes a method that accelerates NeRF training by adaptively allocating network parameters based on geometry details. This paper tackles a practical and important task, enabling faster NeRF training and smaller model size. However, the reviewers raised questions regarding robustness (e.g., sensitivity to the SfM initialization process), presentation (e.g., unstructured related work, unclear performance description), and inferior experimental results. All reviewers unanimously vote for rejection in the absence of a rebuttal.

**Justification For Why Not Higher Score:**

Reviewers have raised reasonable questions about the presentation, experimental and method of the paper, without a rebuttal, all reviewers vote for rejection.

**Justification For Why Not Lower Score:**

N/A

---

### Decision · Program_Chairs · 2024-01-16

Reject